# Is There an Added Value of Quantitative DCE-MRI by Magnetic Resonance Dispersion Imaging for Prostate Cancer Diagnosis?

**DOI:** 10.3390/cancers16132431

**Published:** 2024-07-01

**Authors:** Auke Jager, Jorg R. Oddens, Arnoud W. Postema, Razvan L. Miclea, Ivo G. Schoots, Peet G. T. A. Nooijen, Hans van der Linden, Jelle O. Barentsz, Stijn W. T. P. J. Heijmink, Hessel Wijkstra, Massimo Mischi, Simona Turco

**Affiliations:** 1Department of Urology, Amsterdam UMC, University of Amsterdam, De Boelelaan 1117, 1081 HV Amsterdam, The Netherlands; 2Department of Electrical Engineering, Eindhoven University of Technology, 5612 AP Eindhoven, The Netherlands; 3Leiden University Medical Center, Department of Urology, 2333 ZA Leiden, The Netherlands; 4Department of Radiology and Nuclear Imaging, Maastricht University Medical Centre+, 6229 HX Maastricht, The Netherlands; 5Department of Radiology and Nuclear Medicine, Erasmus University Medical Center, 3015 GD Rotterdam, The Netherlands; 6Department of Radiology, Netherlands Cancer Institute, 1066 CX Amsterdam, The Netherlands; 7Department of Pathology, Jeroen Bosch Hospital, 5223 GZ ‘s-Hertogenbosch, The Netherlands; 8Department of Radiology, Radboud University Nijmegen Medical Center, 6525 GA Nijmegenfi, The Netherlands

**Keywords:** prostate cancer, pharmacokinetic analysis, dynamic constrast-enhanced MRI, multiparametric MRI

## Abstract

**Simple Summary:**

This multicenter, retrospective study assessed the added value of magnetic resonance dispersion imaging (MRDI), a quantitative analysis of dynamic contrast-enhanced MRI (DCE-MRI), alongside standard multiparametric MRI (mpMRI) for detecting clinically significant prostate cancer (csPCa). Seventy-six patients, including fifty-one with csPCa, who underwent mpMRI and radical prostatectomy, were included. Two radiologists evaluated mpMRI, MRDI and a combination of both, with histopathology serving as the reference standard. The study found that MRDI improved inter-observer agreement and enhanced csPCa detection when combined with mpMRI. MRDI enabled the detection of up to 20% more cases compared to mpMRI alone. With the role of DCE-MRI in the context of mpMRI being debated, this study suggests that quantitative analysis of DCE-MRI by MRDI could enhance csPCa detection and reduce variability between observers.

**Abstract:**

In this multicenter, retrospective study, we evaluated the added value of magnetic resonance dispersion imaging (MRDI) to standard multiparametric MRI (mpMRI) for PCa detection. The study included 76 patients, including 51 with clinically significant prostate cancer (csPCa), who underwent radical prostatectomy and had an mpMRI including dynamic contrast-enhanced MRI. Two radiologists performed three separate randomized scorings based on mpMRI, MRDI and mpMRI+MRDI. Radical prostatectomy histopathology was used as the reference standard. Imaging and histopathology were both scored according to the Prostate Imaging-Reporting and Data System V2.0 sector map. Sensitivity and specificity for PCa detection were evaluated for mpMRI, MRDI and mpMRI+MRDI. Inter- and intra-observer variability for both radiologists was evaluated using Cohen’s Kappa. On a per-patient level, sensitivity for csPCa for radiologist 1 (R1) for mpMRI, MRDI and mpMRI+MRDI was 0.94, 0.82 and 0.94, respectively. For the second radiologist (R2), these were 0.78, 0.94 and 0.96. R1 detected 4% additional csPCa cases using MRDI compared to mpMRI, and R2 detected 20% extra csPCa cases using MRDI. Inter-observer agreement was significant only for MRDI (Cohen’s Kappa = 0.4250, *p* = 0.004). The results of this study show the potential of MRDI to improve inter-observer variability and the detection of csPCa.

## 1. Introduction

Prostate cancer (PCa) is the most commonly diagnosed cancer in males, accounting for nearly one in three new cancer diagnoses in the United States in 2024 [1]. In recent years, multiparametric magnetic resonance imaging (mpMRI) has established itself as a reliable diagnostic tool for PCa detection. There is strong evidence (level 1a) supporting the accuracy of mpMRI in detecting clinically significant PCa (csPCa), with sensitivities reaching up to 91% when compared to template biopsy [2]. Several large-scale clinical trials have further confirmed the benefits of incorporating mpMRI into the prostate cancer evaluation process, with MRI-targeted biopsy (MRI-TBx) finding an additional 6.3% to 7.6% csPCa compared to conventional systematic biopsy (SBx) [2,3,4]. As a result, the use of pre-biopsy mpMRI has become a standard practice in many institutions for evaluating patients suspected of having PCa.

Traditionally, the prostate MRI protocol consists of T2-weighted imaging (T2W), diffusion-weighted imaging (DWI) and dynamic contrast-enhanced imaging (DCE). While the role of T2W and DWI sequences is well established, the added value of DCE for PCa detection is currently debated [5]. Biparametric MRI (bpMRI), an alternative to mpMRI that does not include DCE, is gaining popularity due to its reduced imaging time and cost. Currently published data on the diagnostic accuracy of bpMRI show mixed results. Multiple recent meta-analyses comparing bpMRI and mpMRI show no difference in diagnostic accuracy [6,7,8]. However, caution is warranted considering that these data originate from single-center studies, and there are no large, prospective, randomized controlled trials available. Other studies show that DCE MRI does improve sensitivity for csPCa detection [9,10,11]. DCE can play an especially important role in further characterizing PI-RADS 3 lesions located in the peripheral zone (PZ), with Greer et al. finding an odds ratio of 2.0 (*p* = 0.27) for csPCa detection. In the Prostate Imaging—Reporting and Data System (PI-RADS) V2, DCE is considered to be of secondary importance to T2W and DWI [5]. However, the updated PI-RADS V2.1 protocol states that DCE can still be of value for csPCa detection, especially when either the T2W or DWI sequence is of suboptimal quality (e.g., artifacts or inadequate signal-to-noise ratio) [12]. Additionally, the PI-RADS Steering Committee voices their concerns that widespread implementation of bpMRI can lead to missed csPCa cases [12].

In DCE-MRI, a bolus of gadolinium-based contrast agent is administered intravenously during rapid T1-weighted imaging. The contrast flows through the microvasculature, where it is temporarily confined, after which it diffuses into the extracellular space, or “leakage space” [13]. The rate at which inflow and diffusion take place depends on multiple factors related to the microvascular structure [13]. In PCa, angiogenesis causes alternations to this microvascular structure, leading to abnormalities in perfusion and permeability [14]. These abnormalities can be observed on DCE-MRI as early focal contrast enhancement and fast contrast washout [13]. While visual or qualitative assessment is the most commonly performed method for DCE-MRI assessment, it is subjective and susceptible to inter-observer variability [11]. To increase the reproducibility of DCE-MRI, semi-quantitative and quantitative analysis methods have been proposed. Semi-quantitative analysis of the extracted time–intensity curves provides parameters describing tissue enhancement (e.g., peak enhancement, wash-in, wash-out) as a predictor of malignancy [15]. However, these parameters are subject to high interpatient variability and are difficult to generalize due to variations in acquisition protocols and sequences [16,17].

Quantitative analysis of DCE-MRI is based on the quantification of intravascular contrast leakage to the extracellular space. This can be accomplished using compartmental pharmacokinetic modeling devised by Tofts et al. [18]. The Tofts model (TM) describes two main pharmacokinetics (PK) parameters. These parameters can quantify contrast leakage from plasma to tissue and have shown to be significantly increased in cancerous prostate tissue [19]. The major advantage of quantitative DCE-MRI over qualitative and semi-quantitative analysis is that it does not depend on the MRI scanner brand or model, pulse sequence, observer experience or contrast administration protocol [20]. However, the reliability of the parameters is limited by other factors. TM relies on the Arterial Impulse Function (AIF) for PK parameter estimation. The AIF represents the concentration of the contrast agent in the plasma [21]. When evaluating the AIF determination in a multicenter setting, significant variations are found [22,23]. These variations have a considerable impact on PK parameter estimation and therefore limit the reliability of TM [20]. Studies comparing TM for quantitative DCE-MRI to the semi-quantitative and qualitative methods have thus far not shown an improvement in PCa detection [24].

In an effort to overcome the current limitations of TM, novel methods for quantitative analysis for DCE-MRI are being developed [25,26]. One of these methods is magnetic resonance dispersion imaging (MRDI) [27,28,29]. This method quantifies the dispersion of an extravascular contrast agent by the local dispersion parameter κ at each voxel in the prostate. The dispersion parameter κ is highly dependent on the microvascular changes caused by angiogenesis and can therefore be used to construct parametric maps that are suitable for PCa detection [28,29]. Contrary to TM, MRDI does not require AIF determination, thus preventing variation in the parameter estimates caused by AIF inaccuracies [20].

Two previous studies comparing MRDI to whole-gland prostate histopathology have proven the potential of MRDI for PCa detection and localization [28,29], with MRDI outperforming all TM parameters and reaching a sensitivity of 91% [28]. To prove the clinical utility of MRDI, further validation is necessary. The aim of this study is to evaluate the diagnostic potential of quantitative DCE-MRI analysis by MRDI for the detection and localization of csPCa as a separate imaging modality and as an addition to mpMRI, using histopathology from radical prostatectomy specimens as the reference standard.

## 2. Materials and Methods

### 2.1. Patient Population and Data Acquisition

Participants in this study were recruited from three tertiary healthcare centers in the Netherlands within the framework of the Prostate Cancer Molecular Medicine (PCMM) project. Ethical approval for data utilization was granted by the medical ethics review committee of Erasmus MC (Rotterdam, The Netherlands) under the reference number NL32105.078.10. The PCMM project systematically gathered mpMRI and pathology data from men diagnosed with localized PCa, who were scheduled for prostatectomy. Data collection was performed prospectively from 7 February 2011 to 30 June 2015. All study participants provided written informed consent to have data from their medical records used in research.

The adopted pre-biopsy prostate MRI protocol depended on the center of acquisition. Table 1 gives an overview of the MRI acquisition details for each participating center.

### 2.2. MRDI Analysis

Time–intensity curves were obtained from DCE-MRI images for each pixel and converted to concentration–time curves as explained in [28]. Quantitative analysis was performed by fitting each concentration–time curve by the reduced dispersion model, according to a previously described method known as MRDI [27,28,29]. Parametric maps of the local dispersion parameter *κ* were obtained and visualized as color-coded maps using a custom-made software tool (Figure 1a).

### 2.3. MRI and MRDI Scoring

Assessment was performed by two radiologists, R1 and R2, with 9 and 5 years of experience in PIRADS scoring, respectively, who were blinded to the histopathology results. Scoring was performed according to the PIRADS V2.0 prostate sector map (Figure 1b). PIRDS V2.0 was adopted because the development of the protocols and the start of the study occurred before the publication of the updated PIRADS V2.1 [12]. Each radiologist performed three randomized scorings by evaluating mpMRI alone, MRDI maps alone and mpMRI and MRDI in conjunction (mpMRI+MRDI). To reduce the risk of bias, the study protocol dictated a pause of at least 2 weeks between performing the different scoring methods. Moreover, patient numbers and order were randomized for each scoring:**mpMRI**: The scoring was performed according to the PIRADS V2.0 guidelines [5].**MRDI**: MRDI maps were scored from 0 (no lesion) to 5 according to custom guidelines summarized in Table 2. No size criterium is given for scoring MRDI, as it is based on the assessment of angiogenic vascularization within and surrounding the tumor [28,29].**mpMRI+MRDI**: The scoring was performed by integrating the information provided by mpMRI and MRDI, according to the separate scoring models for each modality. In the case of a discrepancy between mpMRI and MRDI scores, the final score was at the radiologist’s discretion.

For all scoring methods, a score ≥3 was considered positive (e.g., suspicious for csPCa), while a score <3 was considered to be not suspicious for the presence of csPCa.

### 2.4. Prostate Histopathology

All patients underwent radical prostatectomy (RP) at their respective institutions, and histopathologic analysis was performed on each prostate specimen after resection. After fixation in formalin, the prostate specimens were cut into slices with a thickness of approximately 4 mm by a pathologist who marked cancer areas on the basis of the microscopic analysis of cellular differentiation (Figure 2). For each patient, at least the index lesion was graded by the pathologist according to the 2005 International Society of Urological Pathology (ISUP) Gleason grading system [30], and the corresponding Gleason score (GS) was noted. Based on the histopathological analysis, each sector in the PIRADS sector map (Figure 1b) was scored in consensus by two uropathologists, who were blinded for MRI and MRDI results, according to the criteria summarized in Table 3.

### 2.5. Evaluation of Diagnostic Performance

The added value of quantitative DCE-MRI analysis by MRDI to the standard mpMRI protocol for csPCa detection was evaluated on a per-patient level, using RP specimen histopathology as the reference standard. A positive mpMRI, MRDI or mpMRI+MRDI (e.g., suspicious for csPCa) was defined as at least one sector scored as 3 or higher. This was considered a true positive when csPCa, defined as any GS ≥ 3 + 4 = 7, was present in the RP histopathology.

Performance of mpMRI and MRDI was also separately evaluated to determine the number of csPCa prostates missed by mpMRI and detected by MRDI and vice versa.

### 2.6. Prostate Cancer Localization

The ability to localize csPCa was assessed on a per-sextant level. Sextants were created by dividing each prostate slice from the PIRADS V2.0 prostate sector map (apex, mid, base) into left and right, thereby creating six areas (sextants). The diagnostic performance of mpMRI and MRDI was evaluated using the RP specimen pathology scoring as the reference standard for the corresponding sextants. For mpMRI and MRDI, a sextant was considered to contain csPCa if at least one sector in the sextant was scored ≥3; for the pathology scoring, this was ≥4. To reduce the influence of mismatching errors, a correction was applied by looking at each sector with csPCa in the pathology and considering the corresponding MRDI/mpMRI/MRDI+mpMRI sector to contain csPCa also when the sector adjacent to it in the same slice or the same sector in an adjacent slice (apex-mid or mid-base) contained csPCa. The correction was applied before aggregating the results for each sextant.

### 2.7. Statistical Analysis

Diagnostic performance was expressed as sensitivity and specificity for both the per-patient and per-sextant analysis. Performance was evaluated for mpMRI, MRDI and MRDI+mpMRI. Statistically significant differences in sensitivity and specificity were evaluated using the McNemar Chi-squared test with Yates’s correction. Discrepancies between mpMRI and MRDI were evaluated on a per-case basis.

Inter- and intra-observer variability for the radiologists were evaluated for each scoring method using Cohen’s Kappa on a per-patient level.

## 3. Results

### 3.1. Patient Population

The PCMM database consisted of 90 patients. After exclusion for insufficient temporal resolution of the DCE exam for MRDI analysis, movement artifacts and missing DICOM files, a total of 76 patients were included for analysis. Table 4 gives an overview of patient characteristics and histopathology findings [31]. Table 5 shows PI-RADS scores for both radiologists.

### 3.2. Diagnostic Performance

Most csPCa lesions were found by both mpMRI and MRDI; however, each technique showed additional value above the other. For R1, 16% (8 out of 51) of csPCa was detected on mpMRI only, and 4% (2 out of 51) of csPCa was detected on MRDI only. For R2, this was 4% (2 out of 51) and 20% (10 out of 51), respectively. These results are also described in Figure 3. Two example cases of mpMRI imaging, MRDI maps and corresponding histopathology are shown in Figure 4. The selected histopathology slice was visually matched to the mpMRI. For the case in (a), both R1 and R2 missed the clinically significant lesion (GS = 3 + 4) on MRDI but found it on mpMRI. For the case in (b), both R1 and R2 missed the clinically significant lesion (GS = 3 + 4) on mpMRI but found it on MRDI.

Table 6 and Table 7 show the sensitivity, specificity, and accuracy for the csPCa detection of each imaging technique on a per-patient and a per-sextant level, respectively. The per-sextant analysis provides an indication of the localization performance. No significant differences were found in the performance of mpMRI, MRDI and mpMRI+MRDI for either of the radiologists.

### 3.3. Per-Patient Discrepancies between Imaging and Pathology

For R1, two (3.9%) csPCa cases were missed by mpMRI only, eight (15.6%) on MRDI only and one (2.0%) on both. R2 missed 10 (19.6%) cases of csPCa on mpMRI only, 2 (3.9%) on MRDI only and 1 (2.0%) on both. Table 8 gives an overview of the missed cases per scoring method and corresponding pathology results. Note that the ISUP > 1 is considered csPCa.

### 3.4. Per-Patient Inter-Observer Variability

Inter-observer agreement between R1 and R2 was fair for mpMRI (κ = 0.1456, *p* = 0.054), moderate for MRDI (κ = 0.4250, *p* = 0.004) and poor for mpMRI+MRDI (κ = −0.0585, *p* = 0.681). Significant inter-observer agreement was only found for MRDI.

Intra-observer agreement for R1 was slight for mpMRI vs. MRDI (κ = 0.1375, *p* = 0.365), perfect for mpMRI vs. mpMRI+MRDI (κ = 1, *p* = 0.000) and slight for MRDI vs. mpMRI+MRDI (κ = 0.1375, *p* = 0.365). For R2, intra-observer agreement was slight for mpMRI vs. MRDI (κ = 0.1582, *p* = 0.340), poor for mpMRI vs. mpMRI+MRDI (κ = −0.0747, *p* = 0.707) and poor for MRDI vs. mpMRI+MRDI (κ = −0.0585, *p* = 0.681). Significant intra-observer agreement was only found for mpMRI vs. mpMRI+MRDI for R1.

## 4. Discussion

This study presents the first assessment of radiologists’ experience with interpreting MRDI in combination with mpMRI in patients undergoing radical prostatectomy (RP). The aim was to determine the additional value of MRDI for the detection of clinically significant prostate cancer (csPCa) compared to the current standard of care, mpMRI. The results of the study suggest that MRDI can be of additional value for PCa diagnosis as there were cases where the radiologist correctly identified csPCa on MRDI, which were not detected on mpMRI.

The sensitivity for csPCa detection varied between the two radiologists, with R1 detecting more csPCa using mpMRI and R2 detecting more csPCa using MRDI. However, for R2, the combined reading of mpMRI and MRDI led to a substantial improvement in csPCa detection compared to mpMRI alone, while R1 did not show any improvement. The differences in added value between the radiologists could be attributed to the challenges of interpreting a new imaging modality, and more extensive training is necessary to fully utilize the potential of MRDI. This is demonstrated by Figure 4a, where both radiologists missed a clear lesion on MRDI. To achieve optimal results, it is important to continuously improve the interpretation and integration of MRDI and mpMRI information.

The low specificity of both imaging modalities on a per-patient level can be attributed to the highly selected patient population in this study. The 25 patients who were deemed negative for csPCa were all treated by RP due to ISUP 1 PCa. ISUP 1 PCa lesions can be visible on mpMRI and are often interpreted as significant lesions, especially in the case of larger ISUP 1 tumors [32]. Bratan et al. evaluated PCa detection by mpMRI using full-mount histopathology as the reference and found that 70% of ISUP 1 tumors larger than 2cc were detected on mpMRI [32]. Therefore, the patient-level specificity in the current study is not representative of the accuracy of the imaging modalities used. This is substantiated by the higher sextant-based specificities.

In this study, we found fair agreement between R1 and R2 for mpMRI (κ = 0.146, *p* = 0.054). The agreement between R1 and R2 for MRDI was substantially higher, reaching significant moderate agreement (κ = 0.425, *p* = 0.004). A possible explanation is the relatively easily (compared to mpMRI) interpretable visualization used in quantitative imaging methods, such as MRDI (see Figure 1a). Furthermore, the accessibility of MRDI could also positively impact the steep learning curve generally associated with mpMRI. This is particularly relevant considering the growing need for skilled radiologists due to the increasing adoption of mpMRI for PCa detection and the increasing incidence of PCa [1,33].

This study has several limitations. First, there is an increased risk for selection bias due to the inclusion of RP patients only, with a larger prevalence of aggressive prostate cancer expected in this population. Based on the current guidelines [34,35], with mpMRI being recommended for pre-biopsy risk stratification of intermediate and high-risk patients, future prospective studies are warranted to validate the proposed approach in biopsy-naive patients, e.g., for biopsy targeting. Second, due to the deformation of the prostate after RP and because of the difference in the slicing angle of pathology and MRI, mismatching errors can occur when correlating imaging to pathology. By dividing the prostate into sextants, we attempted to minimize this error. Lastly, due to its retrospective design and limited sample size, the results cannot yet be extrapolated to general practice. However, this is the first study reporting on a cohort evaluated with MRDI; future prospective studies will need to prove the utility of MRDI in clinical practice. Given the recent advances in other imaging modalities, such as multiparametric ultrasound and micro-ultrasound [36,37], the investigation of multimodal imaging approaches, leveraging mpMRI image fusion with these emerging techniques, is also warranted.

## 5. Conclusions

In this study, we reported the results of the first experience with using MRDI for csPCa detection in patients undergoing RP. The results showed that MRDI has the potential to further increase the diagnostic accuracy of mpMRI, with sensitivity of up to 0.96 achieved with the combination of MRDI and mpMRI. Notably, MRDI could potentially lead to an improvement in detection rate of up to 20% and a reduction in missed csPCa of up to 17% when combined with mpMRI. The quantitative MRDI maps could prove to be especially useful for less-experienced radiologists and for improving inter-observer agreement. However, further validation in a larger, pre-biopsy cohort is necessary before clinical implementation.

## Figures and Tables

**Figure 1 cancers-16-02431-f001:**
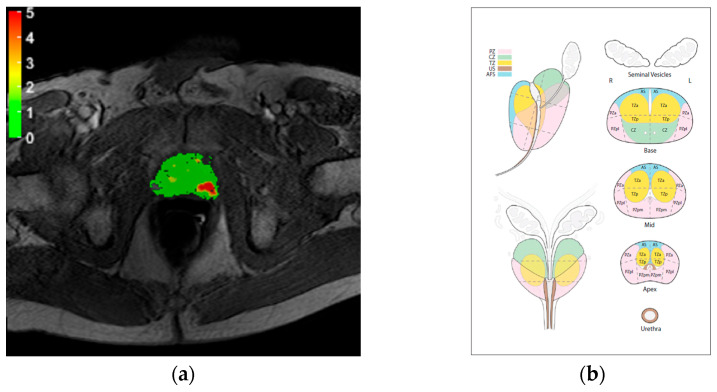
(**a**) Example of MRDI map; (**b**) sector map of PIRADS 2.0 used for scoring.

**Figure 2 cancers-16-02431-f002:**
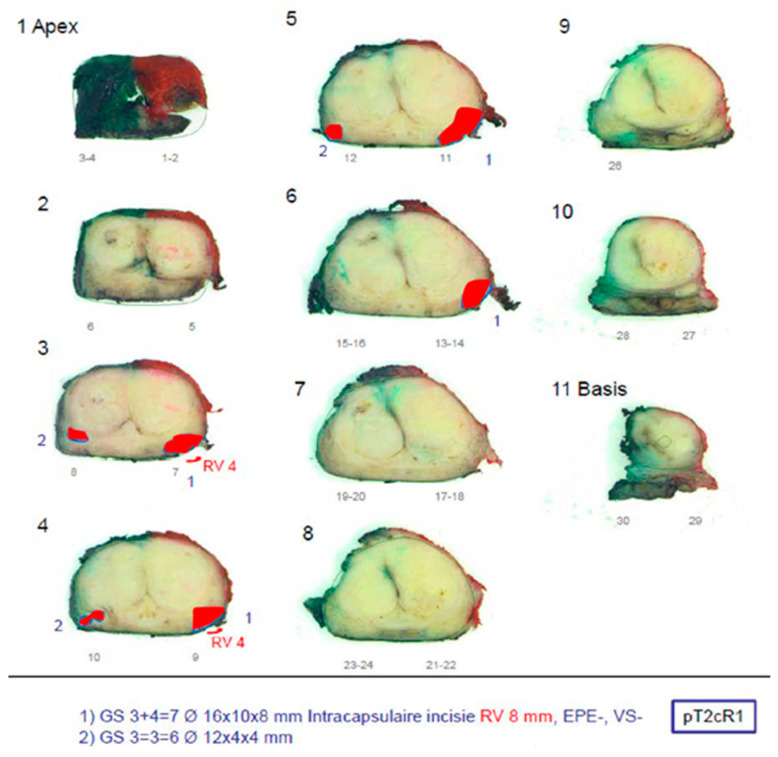
Example of histopathology result.

**Figure 3 cancers-16-02431-f003:**
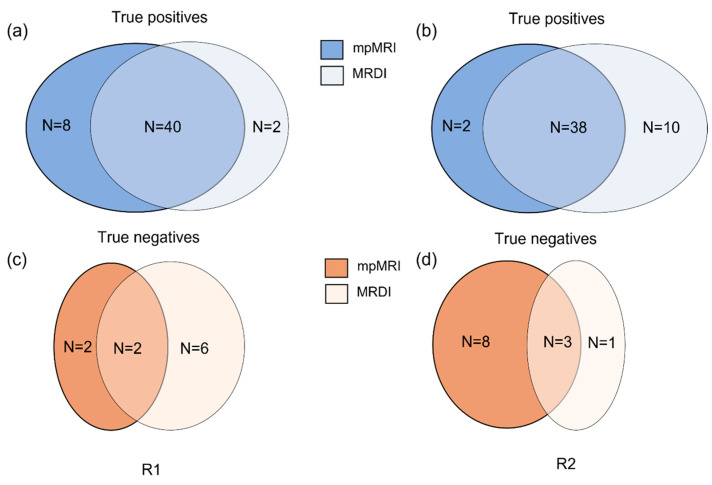
Schematic representation comparing the performance of mpMRI and MRDI. (**a**,**b**) represents the true positives found by mpMRI alone (dark blue), MRDI alone (light blue) and both MRDI and mpMRI (MRDI ∩ mpMRI, mid-tone blue) for radiologists 1 and 2, respectively; (**c**,**d**) represents the true negatives found by mpMRI alone (dark orange), MRDI alone (light orange) and both MRDI and mpMRI (MRDI ∩ mpMRI, mid-tone orange) for radiologists 1 and 2, respectively.

**Figure 4 cancers-16-02431-f004:**
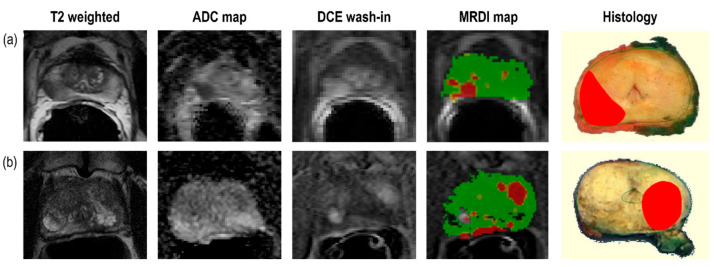
Example cases for two patients showing mpMRI images, MRDI maps and the corresponding visually matched histopathology slice. For both cases, the lesion was diagnosed as Gleason score 4 + 3. For (**a**), both R1 and R2 missed csPCa on MRDI but found it on mpMRI, while for (**b**), both R1 and R2 missed csPCa on mpMRI but found it on MRDI.

**Table 1 cancers-16-02431-t001:** mpMRI acquisition details per center of inclusion. TR = repetition time, TE = echo time.

	T2W			DWI			DCE		
Parameter	Center 1	Center 2	Center 3	Center 1	Center 2	Center 3	Center 1	Center 2	Center 3
TR (ms)	3500–7220	5321–10,233	4000–6050	4000–4800	3429–4498	2500–4200	50	4–5.5	3.85–36
TE (ms)	108	120	99–104	87	67–69	60–90	4	1–2	1.40
Thickness (cm)	3	3	3–4	3–3.6	3	3–4	4–5	6	3–4.5
Width (voxels)	512	512	320–512	136	176	84–160	144	176–256	128–160
Height (voxels)	512	512	320–512	160	176	106–168	192	176–256	128–160
Field strength (Tesla)	1.50	3.00	3.00	1.50	3.00	3.00	1.50	3.00	3.00
Flip angle (degrees)	150	90	117–160	90	90	90	70	8–15	12–14
Endorectal coil (Yes/No)	Yes	Yes	No	Yes	Yes	No	Yes	Yes	No
MRI scanner model	SIEMENS Avanto	Philips Achieva	SIEMENS Skyra/ TrioTim	SIEMENS Avanto	Philips Achieva	SIEMENS Skyra/ TrioTim	SIEMENS Avanto	Philips Achieva	SIEMENS Skyra/ TrioTim
Voxel size (mm)	0.31	0.27	0.31–0.80	1.63	1.03	1.40–2.00	1.67	1.02–2.05	1.50–1.63
MRI sequence	Turbo Spin Echo (TSE)	Turbo Spin Echo (TSE)	Turbo Spin Echo (TSE)	Spin Echo—Echo Planar Imaging (EPI SE)	Spin Echo—Echo Planar Imaging (SE-EPI)	Spin Echo—Echo Planar Imaging (EPI SE)	Spoiled Gradient Echo (FLASH)	Spoiled Gradient Echo (T1-FFE)	Spoiled Gradient Echo (FLASH)
Temporal resolution (s)	-	-	-	-	-	-	3.09–3.12	2.90–3.67	3.31–4.24
Contrast agent	-	-	-	-	-	-	Gadobutrol (0.1 mmol/kg)	Gadoterate meglumine (0.1 mmol/kg)	Gadobutrol (0.1 mmol/kg)

**Table 2 cancers-16-02431-t002:** MRDI map scoring guidelines.

Score	Assessment Category	MRDI Map Features
0	None (benign)	Continuous area with values below 1
1	Very low (clinically significant cancer is highly unlikely to be present)	Continuous area with values between 1 and 2. Non-continuous area with values mostly below 2
2	Low (clinically significant cancer is unlikely to be present)	Continuous area with values between 2 and 3. Non-continuous area with values mostly below 3
3	Intermediate (the presence of clinically significant cancer is equivocal)	Non-continuous area with values between 2 and 4
4	High (clinically significant cancer is likely to be present)	Continuous area with values between 3 and 4. Non-continuous area with values mostly above 4
5	Very high (clinically significant cancer is highly likely to be present)	Continuous area with values above 4

**Table 3 cancers-16-02431-t003:** Guidelines for scoring radical prostatectomy histopathology. Size (%) = tumor volume in the corresponding sector. GS = Gleason score.

Score	Histology	
1	GS ≤ 3 + 3 = 6 Size ≤ 25%	
2	GS ≤ 3 + 3 = 6 Size > 25% & <50&	
3	GS ≤ 3 + 3 = 6 Size ≥ 50%	GS = 3 + 4 = 7 Size ≤ 50%
4	GS > 3 + 4 = 7 Size < 50%	GS = 3 + 4 = 7 Size ≥ 50%
5	GS > 3 + 4 = 7 Size ≥ 50%	

**Table 4 cancers-16-02431-t004:** Demographics and histopathological characteristics of the dataset. pT-stage and ISUP grading is based on radical prostatectomy histopathology. PSA = prostate-specific antigen, ISUP = International Society of Urological Pathology.

**Patient Characteristics**	
Number of patients	76
Age at diagnosis (mean ± std years)	62 ± 6
PSA at biopsy (mean ± std ng/mL)	9 ± 6
Prostate volume (mean ± std mL)	44 ± 18
**pT-stage, *n* (%)**	
T2ab	19 (25)
T2c	32 (42)
T3	25 (33)
**ISUP grade group [31]** **, *n* (%)**	
1	25 (33)
2	27 (36)
3	15 (20)
4	4 (5)
5	5 (6)

**Table 5 cancers-16-02431-t005:** PI-RADS scores for radiologists R1 and R2 and corresponding histopathology results. PI-RADS = Prostate Imaging-Reporting and Data System, csPCa = clinically significant prostate cancer.

R1					R2				
PI-RADS	N	%	N csPCa	% csPCa	PI-RADS	N	%	N csPCa	% csPCa
1	2	2.6	1	50.0	1	0	0.0	0	0.0
2	5	6.6	2	40.0	2	22	28.9	11	50.0
3	13	17.1	8	61.5	3	16	21.1	10	62.5
4	27	35.5	19	70.4	4	23	30.3	16	69.6
5	29	38.2	21	72.4	5	15	19.7	14	93.3
Total	76	100.0	51		Total	76	100.0	51	

**Table 6 cancers-16-02431-t006:** Diagnostic performance in terms of sensitivity, specificity and accuracy on a patient level. TP = true positive, TN = true negative, FN = false negative, FP = false positive, N = ground-truth negative (TN + FP), P = ground-truth positive (TP + FN).

	Sensitivity (TP/P)			Specificity (TN/N)			Accuracy (TN + TP/N + P)		
Radiologist	mpMRI	MRDI	mpMRI+ MRDI	mpMRI	MRDI	mpMRI+ MRDI	mpMRI	MRDI	mpMRI+ MRDI
R1	0.94 (48/51)	0.82 (42/51)	0.94 (48/51)	0.16 (4/25)	0.32 (8/25)	0.16 (4/25)	0.68 (52/76)	0.66 (50/76)	0.68 (52/76)
R2	0.78 (40/51)	0.94 (48/51)	0.96 (49/51)	0.68 (17/25)	0.16 (4/25)	0.04 (1/25)	0.67 (51/76)	0.68 (52/76)	0.66 (50/76)

**Table 7 cancers-16-02431-t007:** Diagnostic performance in terms of sensitivity, specificity and accuracy on a sextant level. TP = true positive, TN = true negative, FN = false negative, FP = false positive, N = ground-truth negative (TN + FP), P = ground-truth positive (TP + FN).

	Sensitivity (TP/P)			Specificity (TN/N)			Accuracy (TN + TP/N + P)		
Radiologist	mpMRI	MRDI	mpMRI+ MRDI	mpMRI	MRDI	mpMRI+ MRDI	mpMRI	MRDI	mpMRI+ MRDI
R1	0.81 (103/127)	0.56 (71/127)	0.81 (103/127)	0.85 (279/329)	0.83 (273/329)	0.85 (279/329)	0.84 (382/456)	0.75 (344/456)	0.84 (382/456)
R2	0.51 (65/127)	0.54 (69/127)	0.61 (77/127)	0.92 (302/329)	0.84 (276/329)	0.85 (281/329)	0.80 (367/456)	0.76 (345/456)	0.79 (359/456)

**Table 8 cancers-16-02431-t008:** Diagnostic performance in terms of sensitivity and specificity on a patient level. TP = true positive, TN = true negative, FN = false negative, FP = false positive, N = ground-truth negative (TN + FP), P = ground-truth positive (TP + FN).

Number of Missed csPCa						
	R1			R2		
ISUP	mpMRI Only	MRDI Only	Missed by Both	mpMRI Only	MRDI Only	Missed by Both
**2**	1/27	5/27	1/27	5/27	0/27	1/27
**3**	1/15	2/15	0/15	4/15	2/15	0/15
**4**	0/4	0/4	0/4	1/4	0/4	0/4
**5**	0/5	1/5	0/5	0/5	0/5	0/5
**Total**	2/51	8/51	1/51	10/51	2/51	1/51

## Data Availability

The original contributions presented in the study are included in the article; further inquiries can be directed to the corresponding author.

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
