# Peer review of "Is There an Added Value of Quantitative DCE-MRI by Magnetic Resonance Dispersion Imaging for Prostate Cancer Diagnosis?"

_cancers, 2024, doi:10.3390/cancers16132431_

Round 1

Reviewer 1 Report

Comments and Suggestions for Authors

The authors conducted an interesting retrospective multicentre study to investigate a quantitative method of dynamic contrast-enhanced MRI analysis to understand whether it is able to give advantages in terms of specificity and sensitivity in detecting clinically significant prostate cancer csPCA on prostate MRI. The quantitative method analysed is called magnetic resonance dispersion imaging (MRDI).

This is a useful clinical and diagnostic target for an imaging modality that was initially used only for staging and has now been confirmed as an essential tool in the early stages of prostate cancer diagnosis.

For this objective, 76 patients were selected from the Prostate Cancer Molecular Medicine Project, all patients with known prostate cancer eligible for prostatectomy for whom MRI images were available with the possibility of applying MRDI.

In this sense, the study is retrospective. However, two independent radiologists were used who were unaware of the already known clinical correlation between the patients and the presence of prostate cancer confirmed by histological studies of the prostatectomies to which the above patients had been subjected. In this sense, this choice reduced the risk of selection bias.

The radiologists had to identify suspicious areas of prostate cancer on the MR images by comparing the images first with the classic mpMRI weightings alone, then with MRDI, and finally with both methods.
The statistical analysis of the method is adequate and correct, and it shows a significant improvement in specificity and sensitivity when using standard mpMRI weightings combined with MRDI.

The article is well written and structured. I agree with the limitations that the authors self-critically highlight in the discussion. However, despite the small sample size and the retrospective analysis, the originality of the method analysed to identify prostate cancer and the results obtained make the article publishable, suggesting the extension of the study to a multicentre or the consideration of a prospective study.

Author Response

[Comment] The authors conducted an interesting retrospective multicentre study to investigate a quantitative method of dynamic contrast-enhanced MRI analysis to understand whether it is able to give advantages in terms of specificity and sensitivity in detecting clinically significant prostate cancer csPCA on prostate MRI. The quantitative method analysed is called magnetic resonance dispersion imaging (MRDI).

This is a useful clinical and diagnostic target for an imaging modality that was initially used only for staging and has now been confirmed as an essential tool in the early stages of prostate cancer diagnosis.

For this objective, 76 patients were selected from the Prostate Cancer Molecular Medicine Project, all patients with known prostate cancer eligible for prostatectomy for whom MRI images were available with the possibility of applying MRDI.

In this sense, the study is retrospective. However, two independent radiologists were used who were unaware of the already known clinical correlation between the patients and the presence of prostate cancer confirmed by histological studies of the prostatectomies to which the above patients had been subjected. In this sense, this choice reduced the risk of selection bias.

The radiologists had to identify suspicious areas of prostate cancer on the MR images by comparing the images first with the classic mpMRI weightings alone, then with MRDI, and finally with both methods.
The statistical analysis of the method is adequate and correct, and it shows a significant improvement in specificity and sensitivity when using standard mpMRI weightings combined with MRDI.

The article is well written and structured. I agree with the limitations that the authors self-critically highlight in the discussion. However, despite the small sample size and the retrospective analysis, the originality of the method analysed to identify prostate cancer and the results obtained make the article publishable, suggesting the extension of the study to a multicentre or the consideration of a prospective study.

[Response]

We would like to thank the reviewer for the appreciation of our study, which encourage further research in this direction.

Reviewer 2 Report

Comments and Suggestions for Authors

This work addresses prostate cancer detection using multiparametric magnetic resonance imaging (mpMRI). The hypothesis is that magnetic resonance dispersion imaging (MRDI) can improve the diagnostic accuracy of standard mpMRI for detecting clinically significant prostate cancer (csPCa).

I find that the manuscript is well-structured and the contributions are clearly articulated. The primary contribution, that MRDI enhances the diagnostic accuracy and inter-observer agreement of standard mpMRI for detecting csPCa, is evident.

However, the conclusions are overly brief. Given the significant findings presented, the conclusions should provide a more comprehensive summary, particularly in quantitative terms. The conclusions section should be expanded to include a detailed summary of the key findings. This should encompass specific quantitative results, such as sensitivity and specificity improvements observed with the addition of MRDI.

Ensure that the methodology section provides sufficient detail to allow reproducibility. This includes specifics on the MRDI implementation and any variations in the mpMRI protocols used across different centers.

While limitations are briefly mentioned, a more in-depth discussion is warranted. This should include potential biases, such as the selection bias due to the inclusion of only radical prostatectomy patients, and the implications of these limitations on the study's findings. Suggest concrete future research directions based on your findings. This could include prospective studies to validate MRDI in larger, more diverse patient cohorts or exploring the integration of MRDI with other emerging imaging modalities.

Please, address the above recommendation into the manuscript.

Author Response

Response 0: 

We would like to express our gratitude for your time and effort for the review of our manuscript. We highly appreciated your constructive and insightful feedback that substantially improved our manuscript. We hope that the information we have added and our responses meet your requirements. Please find below our detailed replies to the comments along with the changes implemented in the manuscript. All changes have also been tracked/highlighted in the revised manuscript.

====================================

Color code used in this response letter:

Black: Original comments to Authors

Blue: Authors’ response

Gray: Changes in the manuscript

====================================

Comment 1: 

This work addresses prostate cancer detection using multiparametric magnetic resonance imaging (mpMRI). The hypothesis is that magnetic resonance dispersion imaging (MRDI) can improve the diagnostic accuracy of standard mpMRI for detecting clinically significant prostate cancer (csPCa).

I find that the manuscript is well-structured and the contributions are clearly articulated. The primary contribution, that MRDI enhances the diagnostic accuracy and inter-observer agreement of standard mpMRI for detecting csPCa, is evident.

Response 1: 

We would like to thank the reviewer for the positive comments.

Comment 2: 

However, the conclusions are overly brief. Given the significant findings presented, the conclusions should provide a more comprehensive summary, particularly in quantitative terms. The conclusions section should be expanded to include a detailed summary of the key findings. This should encompass specific quantitative results, such as sensitivity and specificity improvements observed with the addition of MRDI.

Response 2:

We would like to thank the reviewer for underlying the significance of our findings. We agree that this should be highlighted better in the conclusions. Accordingly, we have extended this section with the following:

“[...], with sensitivity of up to 0.96 achieved with the combination of MRDI and mpMRI. Notably, MRDI could potentially lead to an improvement in detection rate of up to 20% and a reduction in missed csPCa of up to 17% when combined with mpMRI.”

Comment 3

Ensure that the methodology section provides sufficient detail to allow reproducibility. This includes specifics on the MRDI implementation and any variations in the mpMRI protocols used across different centers.

We agree with the reviewer that sufficient detail is necessary to allow for reproducibility. In response, Table 1 provides all the details on the mpMRI protocols used across the three centers, to which we also added the used MRI sequence, as suggested by the editor. Additionally, references 27-29 provide all the details for implementation of MRDI. We are also currently working on making the MRDI analysis code available open source.

Comment 4

While limitations are briefly mentioned, a more in-depth discussion is warranted. This should include potential biases, such as the selection bias due to the inclusion of only radical prostatectomy patients, and the implications of these limitations on the study's findings. Suggest concrete future research directions based on your findings. This could include prospective studies to validate MRDI in larger, more diverse patient cohorts or exploring the integration of MRDI with other emerging imaging modalities

We thank the reviewer for the suggestion. Accordingly, we added the following sentences to the discussion:

“This study has several limitations. First, there is an increased risk for selection bias due to the inclusion of RP patients only, with a larger prevalence of aggressive prostate cancer expected in this population. Based on the current guidelines [34, 35], with mpMRI being recommended for pre-biopsy risk-stratification of intermediate and high-risk patients, future prospective studies are warranted to validate the proposed approach in biopsy-naive patients, e.g., for biopsy targeting.”

“Given the recent advances of other imaging modalities, such as multiparametric ultrasound and micro-ultrasound [36, 37], the investigation of multimodal imaging approaches, leveraging mpMRI image fusion with these emerging techniques, is also warranted.”

New references:

  1. Wei, J.T.; Barocas, D.; Carlsson, S.; Coakley, F.; Eggener, S.; Etzioni, R.; Fine, S.W.; Han, M.; Kim, S.K.; Kirkby, E.; et al. Early Detection of Prostate Cancer: AUA/SUO Guideline Part II: Considerations for a Prostate Biopsy. J. Urol. 2023, doi:10.1097/JU.0000000000003492.
  2. EAU Guidelines on Prostate Cancer - DIAGNOSTIC EVALUATION - Uroweb Available online: https://uroweb.org/guidelines/prostate-cancer/chapter/diagnostic-evaluation (accessed on 19 June 2024).
  3. Basso Dias, A.; Ghai, S. Micro-Ultrasound: Current Role in Prostate Cancer Diagnosis and Future Possibilities. Cancers 2023, 15, 1280, doi:10.3390/cancers15041280.
  4. Wildeboer, R.R.; Mannaerts, C.K.; van Sloun, R.J.G.; Budäus, L.; Tilki, D.; Wijkstra, H.; Salomon, G.; Mischi, M. Automated Multiparametric Localization of Prostate Cancer Based on B-Mode, Shear-Wave Elastography, and Contrast-Enhanced Ultrasound Radiomics. Eur. Radiol. 2020, 30, 806–815, doi:10.1007/s00330-019-06436-w.